# Favorable Vasomotor Function after Drug-Coated Balloon-Only Angioplasty of De Novo Native Coronary Artery Lesions

**DOI:** 10.3390/jcm11020299

**Published:** 2022-01-07

**Authors:** Sunwon Kim, Jong-Seok Lee, Yong-Hyun Kim, Jin-Seok Kim, Sang-Yup Lim, Seong Hwan Kim, Minjung Kim, Jeong-Cheon Ahn, Woo-Hyuk Song

**Affiliations:** Cardiovascular Center, Korea University Ansan Hospital, Ansan-si 15355, Korea; sunwon11@hanmail.net (S.K.); tntwkakclavy@gmail.com (J.-S.L.); xkyhx@hanmail.net (Y.-H.K.); heartmania@unitel.co.kr (J.-S.K.); vnlover@hanmail.net (S.-Y.L.); ny0021@medimail.co.kr (S.H.K.); kminj1184@naver.com (M.K.); hhansin@korea.ac.kr (J.-C.A.)

**Keywords:** drug-coated balloon, de novo lesion, vasospasm, coronary vasomotor function

## Abstract

Balloon-injured coronary segments are known to harbor abnormal vasomotion. We evaluated whether de novo coronary lesions treated using drug-coated balloon (DCB) are prone to vasospasm and how they respond to ergonovine and nitrate. Among 132 DCB angioplasty recipients followed, 89 patients underwent ergonovine provocation test at 6–9 months follow-up. Within-subject ergonovine- and nitrate-induced diameter changes were compared among three different sites: DCB-treated vs. angiographically normal vs. segment showing prominent vasoreactivity (spastic). No patient experienced clinically refractory vasospastic angina or symptom-driven revascularization during follow-up. Ergonovine induced vasospasm in seven patients; all were multifocal spasm either involving (*n* = 2) or rather sparing DCB-treated segments (*n* = 5). None showed focal spasm that exclusively involved DCB-treated lesions. Among 27 patients with vasospastic features, DCB-treated segments showed less vasoconstriction than spastic counterparts (*p* < 0.001). A total of 110 DCB-treated lesions were analyzed to assess vasomotor function. Vasomotor function, defined as a combined constrictor and dilator response, was comparable between DCB-treated and angiographically normal segments (*p* = 0.173), while significant differences were observed against spastic counterparts (*p* < 0.001). In our study, DCB-treated lesions were not particularly vulnerable to vasospasm and were found to have vasomotor function similar to angiographically normal segments, supporting safety of DCB-only strategy in treating de novo native coronary lesions.

## 1. Introduction

Drug-coated balloon (DCB) angioplasty is an emerging treatment approach for obstructive coronary artery disease (CAD). It has demonstrated its safety and efficacy in the treatment of in-stent restenosis and small vessel CAD and, currently, many on-going trials are exploring its value in a broader range of lesions [1,2,3]. Theoretically, the short-term anti-proliferative drug action of DCBs promotes arterial reparative process, allowing early re-endothelialization and vessel healing [4]. Further, the absence of potential inflammatory sources, such as polymer or vessel-caging metallic material, reduces the risk of maladaptive healing responsible for restenosis and thrombosis and allows the vessels to freely constrict and dilate [4,5,6]. As positive consequences, recent studies reported that successful DCB angioplasty could induce positive vessel remodeling and even plaque regression [7,8].

Percutaneous transluminal coronary angioplasty (PTCA), including plain old balloon angioplasty (POBA) or stenting, unavoidably results in significant vascular injury, leading to a variety of deleterious consequences, such as dissection, thrombus formation, impairment of physiologic vasomotion, and restenosis [9]. In particular, evidence in the pre-stent PTCA era repeatedly shows the presence of abnormal vasomotion at the balloon-injured segment, resulting in recurrent acute coronary syndrome (ACS) and even death [10,11,12,13]. DCB angioplasty for de novo native coronary lesions is fundamentally the same procedure that was performed in the pre-stent era, except for the antiproliferative drug being delivered directly to the vessel wall. However, no previous study has evaluated whether a coronary segment that had been subjected to DCB angioplasty and left unscaffolded is prone to vasospasm. In addition, data regarding the coronary vasomotor function after DCB angioplasty are lacking. This study sought to evaluate whether DCB-treated lesions are more vulnerable to vasospasm and how these segments respond to ergonovine and isosorbide dinitrate in comparison with angiographically normal and other untreated segments showing prominent constrictor response to ergonovine at 6–9 months after the index procedure.

## 2. Materials and Methods

### 2.1. Population and DCB Angioplasty Procedure

This prospective all-comers study evaluated all patients with significant CAD who received percutaneous coronary intervention (PCI) with DCB-only angioplasty, concomitant DCB angioplasty and POBA, or concomitant DCB angioplasty and drug-eluting stent (DES) implantation for de novo lesions in the native coronary arteries at the Korea University Ansan Hospital between July 2019 and December 2020. The patients with a life expectancy shorter than 2 years and those who refused to provide their consent were excluded. DCB angioplasty was performed in accordance with the recent recommendations [1,2,14]. Specifically, aggressive lesion predilation (balloon-to-artery ratio: 0.8 to 1.2) using either a plain balloon (78.8%) or a scoring balloon (21.2%) (NSE Alpha, Goodman Co., Ltd., Nagoya, Japan) was performed. Dissections of Types A to C were left unstented, unless flow limitation developed. Bail-out stenting was performed for coronary dissections of Type D or higher. The DCB (SeQuent Please or SeQuent Please NEO, B. Braun Melsungen AG, Melsungen, Germany) was inflated for at least 30 s and mostly over 60 s at the discretion of the interventionist. After PCI, all patients received dual antiplatelet treatment for 6 months, unless significant bleeding events occurred. The participants were followed up clinically every 3 months and recommended to undergo 6- to 9-months follow-up coronary angiography for reassessment of the DCB-treated lesions. Occurrence of new-onset angina and unplanned, symptom-driven clinic visit or rehospitalization was monitored.

### 2.2. Ergonovine Provocation Test

The presence of significant vasospasm and coronary artery vasomotor function were assessed via intracoronary injection of ergonovine and isosorbide dinitrate [14]. All participants were educated not to take oral nitrates, calcium channel blockers, β-blockers, angiotensin-converting enzyme inhibitors, angiotensin II receptor blockers, and other vasoactive medicines (nicorandil and molsidomine) at least 24 h before follow-up coronary angiography. Ergonovine was administered in incremental doses of 20 μg (E1), 40 μg (E2), and 80 μg (E3) into the left coronary artery and 10 μg (E1), 20 μg (E2), and 40 μg (E3) into the right coronary artery for 20–30 s at 2–3-min intervals under continuous electrocardiographic monitoring. Further ergonovine infusion was withheld when the patients developed significant vasospasm or severe chest pain or when ST-segment changes or a clear reduction in coronary blood flow occurred. Angiography was repeated after intracoronary administration of 200–400 ug of isosorbide dinitrate while the coronary artery was maximally dilated [14]. The ergonovine provocation test was not performed when the patients had significant restenosis (>50%) or signs of abnormal healing in the previously DCB-treated segments, lesions treated with a hybrid strategy (DES implantation + DCB angioplasty), diffuse severe three-vessel disease in which an adequate control segment could not be found, documented spasm owing to other causes, chronic renal insufficiency (creatinine level of >1.2 mg/dL) or low left ventricular ejection fraction (<5–40%) according to the discretion of the physician, short left main trunk or coronary ostial lesion in which an appropriate drug delivery to lesion could not be guaranteed, and other significant medical conditions.

### 2.3. Coronary Spasm, Vasomotor Function, and Quantitative Coronary Angiography (QCA)

Coronary spasm was defined as transient, total, or near-total occlusion (>90%) of a coronary artery as recommended [14]. Spasm was classified as focal (vasoconstriction within the confines of one isolated coronary segment) and diffuse or multifocal (vasoconstriction of two or more segments) [14]. Two expert researchers who were blinded to the study purpose measured the QCA parameters (Axiom-Artis, Siemens, Germany or Philips Azurion, Philips Healthcare, the Netherlands). The QCA parameters were measured when the coronary arteries were considered maximally constricted or dilated by the drugs. The percent diameter change was measured at the site showing the greatest differences and calculated as follows:-Constrictor response (%): (Minimal lumen diameter (MLD) after ergonovine administration—MLD after nitrate administration)/MLD after nitrate administration × 100-Dilator response (%): (MLD after nitrate administration—MLD at baseline)/MLD at baseline × 100

The reference vessel diameter (RVD), measured at the proximal and distal portions of each segment of interest, was used to calculate the diameter stenosis (DS) for quantitative comparative analysis. Segments with a percent change of >50% after the ergonovine provocation test were considered to have significant vasoconstriction, and patients who had at least one or more coronary segments showing significant vasoconstriction (>50%) were considered to have vasospastic features.

### 2.4. Comparison of the Vasomotor Function between Ergonovine and Isosorbide Dinitrate

Vasomotor response to ergonovine and isosorbide dinitrate was compared among three different segments: DCB-treated segments, angiographically normal arterial segments (control), and segments showing prominent vasoconstrictive response to ergonovine (spastic) (Figure 1: also refer to Appendix A). The control segment was defined as an arterial segment with smooth vessel contour without significant angiographic stenosis (DS: <30%) or calcification, which was mostly found in the proximal segment of the unintervened arteries or in the intervened arteries at least 10–20 mm apart from the DCB-treated lesion. The spastic segment was defined as an untreated segment having RVD of at least greater than 2 mm and showing the prominent vasoconstrictive response to ergonovine. It included mildly diseased segments with DS of <50% (mean DS: 17.57 ± 1.00%). The arterial segments were selected by two independent observers who were blinded to the study purpose. Within-subject differences in the serially assessed ergonovine- and nitrate-induced percent diameter changes were analyzed.

### 2.5. Statistical Analysis

Continuous variables are expressed as means ± standard deviations and dichotomous variables as counts and percentages. Differences in MLD between baseline and after drug administration were analyzed using a paired *t*-test. Ergonovine-induced percent diameter changes (constrictor response) between DCB-treated vs. spastic segments were compared using paired *t*-test or Wilcoxon signed-rank test based on normality test. Differences in combined constrictor and dilator responses across the three segments were analyzed using general linear model two-way repeated measures analysis of variance (ANOVA). Repeated measures ANOVA incorporated the Bonferroni test for pair-wise comparison. Statistical analyses were performed using the SPSS software (version 20.0; IBM Corp., Armonk, NY, USA). Statistical significance was set at *p*-values of <0.05.

## 3. Results

### 3.1. Study Population and Clinical Events during Follow-Up

A total of 132 patients who underwent DCB angioplasty for de novo native coronary lesions were analyzed. Clinical characteristics of the study population is described in Appendix A. During hospitalization, no patients experienced recurrent coronary events related to post-DCB angioplasty vasospasm. All patients received either statin monotherapy (14.6%) or statin/ezetimibe combination therapy (85.4%), and 83.4% of the patients achieved a target low-density lipoprotein cholesterol level goal of <70 mg/mL at 3–6 months of follow-up.

During the median follow-up period of 18.5 months, two patients underwent repeated PCI due to recurrent ACS; one patient presented with unstable angina due to a de novo stenotic lesion in the treated vessel (2 months post-PCI); and one patient presented with non-ST elevation myocardial infarction due to post-POBA restenosis of the proximal right coronary artery but whose DCB-treated proximal left anterior descending artery remained patent. Albeit asymptomatic, three patients underwent redo DCB angioplasty owing to post-DCB angioplasty restenosis and one patient underwent DCB angioplasty owing to a newly-developed lesion at the follow-up angiography. One patient complained a sublingual nitroglycerin-responsive, alcohol-related nocturnal chest pain during the follow-up. His symptom was controlled by prescribing calcium channel blocker and oral nitrates, and he later showed ergonovine-induced multifocal vasospasm at the follow-up angiography (Appendix A). In summary, none of the patients had ACS or clinically refractory vasospastic angina attributable to the DCB-treated lesions.

### 3.2. Occurrence of Ergonovine-Induced Vasospasm

Among 132 enrollees, patients who underwent 6–9 months of follow-up invasive coronary angiography and those excluded for the ergonovine provocation test are summarized in the CONSORT flow diagram in Figure 2. The clinical and lesional characteristics, and procedural details of 89 patients who underwent the ergonovine provocation test are summarized in Table 1. Ergonovine induced vasospasm in seven patients. There were no cases of focal vasospasm that developed exclusively in the DCB-treated segment. In 5 cases, the DCB-treated lesions remained patent, while the other arterial segments developed total or subtotal occlusion by ergonovine (Figure 3A). A total of two cases were multifocal spasm that involved both DCB-treated segments and other arterial territories (Figure 3B).

In seven patients with documented vasospasm, RVD and DS were similar between eight DCB-treated segments and their spastic counterparts [DCB-treated vs. spastic: RVD (mm): 2.98 ± 0.56 vs. 2.61 ± 0.37, *p* = 0.069; DS (%): 27.64 ± 10.07 vs. 31.46 ± 13.64, *p* = 0.498]. However, ergonovine-induced constrictor response (%) was significantly lower in the DCB-treated segments than in the spastic segments (DCB-treated vs. spastic: −47.4 ± 30.4% vs. −84.5 ± 14.7%, *p* = 0.017, Figure 3C). The prevalence of segments showing significant vasoconstriction (>50%) was as follows: control vs. DCB-treated vs. spastic segments: 0% vs. 8.5% vs. 29.2%, respectively. Further, in the analysis of 27 patients with vasospastic features (30 DCB-treated lesions), DCB-treated segments showed less constrictor response compared to the spastic counterparts (DCB-treated vs. spastic: constrictor response (%): −40.0 ± 21.1 vs. −66.0 ± 17.8, *p* < 0.001, Figure 3D), despite their similarities in RVD and DS (DCB-treated vs. spastic: RVD (mm): 2.83 ± 0.49 vs. 2.65 ± 0.44, *p* = 0.109; DS (%): 24.19 ± 0.97 vs. 24.66 ± 12.53, *p* = 0.845).

### 3.3. Segment-Wise Vasomotor Response to Ergonovine and Isosorbide Dinitrate

A total of 274 segments (82 control, 110 DCB-treated, and 82 spastic segments) from 82 patients were analyzed after excluding those from seven vasospasm cases. Angiographic characteristics of the analyzed segments are presented in Table 2. Control segments were significantly greater RVD and lower DS than those of DCB-treated and spastic segments (control vs. DCB-treated vs. spastic: RVD (mm): 3.12 ± 0.60 vs. 2.72 ± 0.41 vs. 2.70 ± 0.48, *p* < 0.001; DS (%): 12.72 ± 7.14 vs. 24.21 ± 8.59 vs. 17.56 ± 10.45, *p* < 0.001). DCB-treated segments showed similar RVD (*p* > 0.999), but greater DS compared to spastic segments (*p* < 0.001).

After ergonovine administration, most segments showed significant constriction [MLD (mm): baseline vs. ergonovine: control: 2.54 ± 0.57 vs. 2.20 ± 0.57, *p* < 0.001; DCB-treated: 1.90 ± 0.42 vs. 1.58 ± 0.41, *p* < 0.001; spastic: 1.94 ± 0.46 vs. 1.37± 0.44, *p* < 0.001, Appendix A]. Nitrate induced significant vasodilation in most segments (MLD (mm): baseline vs. nitrate: control: 2.54 ± 0.57 vs. 2.73 ± 0.61, *p* < 0.001; DCB-treated: 1.90 ± 0.42 vs. 2.06 ± 0.40, *p* < 0.001; spastic: 1.94 ± 0.46 vs. 2.22 ± 0.47, *p* < 0.001, Appendix A).

Intriguingly, some DCB-treated segments were found to be unaffected by ergonovine. 35.5% (39 out of 110) of DCB-treated segments had a diameter of at least 20% greater than the adjacent arterial portions after ergonovine administration (Figure 4). These constrictor-resistant segments were frequently found at the most severely narrowed, culprit area or at sites where post-DCB angioplasty dissection developed, implicating a balloon injury mechanism behind this phenomenon.

### 3.4. Comparison of the Comprehensive Vasomotor Function

Vasomotor function, as comprehensively assessed as a combined constrictor and dilator response, was analyzed using repeated measure ANOVA. Within-subject difference in the ergonovine- and nitrate-induced serial percent diameter changes was statistically significant across the three segments (ergonovine vs. nitrate: control: −18.78 ± 10.45 vs. 7.79 ± 8.48; DCB-treated: −22.95 ± 15.57 vs. 9.36 ± 10.73; spastic: −37.03 ± 17.56 vs. 15.55 ± 15.19, *p* < 0.001, Figure 5). The comprehensive constrictor and dilator responses were similar between the DCB-treated and control segments (*p* = 0.173), while these were significantly greater in the spastic segments than in the DCB-treated and control segments (control vs. spastic: *p* < 0.001; DCB-treated vs. spastic: *p* < 0.001, Figure 5).

## 4. Discussion

Coronary artery spasm is not only a functional abnormality of the coronary artery, causing transient, reversible myocardial ischemia, but it indeed plays a role in the pathogenesis of ACS [15]. A previous autopsy study has demonstrated that abnormal vasoconstriction causes endothelial derangement and rupture of the fibrous cap, leading to volcano-like protrusion of the plaque core and subsequent atherothrombosis [16]. The coronary segments showing vasospasm were also revealed to be particularly susceptible to progression of atherosclerosis [17]. As mentioned above, post-PTCA vasospasm is a devastating complication affecting revascularization success [10,11,12,13]. These evidences support that coronary vasomotion is closely involved in various stages of CAD. In our study, the DCB-treated segment did not harbor an increased risk of vasospasm, either clinically or by ergonovine provocation. No case showed vasospasm that developed exclusively in the DCB-treated segments, and these segments had less vasoconstriction than their spastic counterparts, implicating that DCB-treated lesion is less likely to be spasm culprit. Our findings support that DCB angioplasty can be safely performed in patients with vasospastic features.

The introduction of metallic stents was a major breakthrough in overcoming the limitations of PTCA, such as abrupt vessel closure, acute recoil, dissection, and vasospasm. However, permanent vessel caging by metallic implants constitutes barriers to physiological vasomotion and expansile vessel remodeling. Further, DESs implantation produces prolonged endothelial dysfunction in stented artery [6] and, as a result, the proximal and distal adjacent segments of DESs were shown to have dysfunctional vasomotion [18]. Unlike atherosclerosis of the native coronary arteries that build up through decades, the development of neoatherosclerosis in stented arteries was found to be significantly accelerated [5,6]. Incremental plaque growth in a caged artery results in greater lumen loss, which upon exceeding a critical limit leads to clinical events [6]. Consequently, although stent design iterations have improved acute clinical results, a persistent annualized event rate of 2–3% without a plateau effect is observed after coronary stenting regardless of the stent type, including contemporary DESs [19]. DCB angioplasty is expected to help rid patients of adverse events related to permanent coronary implants, which is based on the assumption that the absence of metallic caging would facilitate restoration of vasomotion and ultimately induce luminal enlargement and reduction of plaque burden [7,8]. This study for the first time demonstrated that the DCB strategy, completely devoid of vessel-caging material, could allow restoration of the coronary vasomotor function as early as 6 months after angioplasty, achieving levels comparable to those of angiographically healthier, untreated segments.

The mechanisms underlying the observed favorable findings may be multifactorial. Ergonovine, an ergot derivative, exerts vasoconstrictive effects by stimulating alpha-adrenergic and serotonin receptors [20] and inhibiting endothelial-derived relaxation factor release [21]. Given that the vasoconstrictor action of ergonovine is reinforced in the absence of a functional endothelium [21], our findings in part support that DCB angioplasty, through stent-less balloon-based short-term drug delivery, allows the endothelium to rapidly recover its functional integrity. Further, aggressive and repeated ballooning during DCB angioplasty unavoidably causes a significant injury to the smooth muscle layer and thus might render the vessels incapable of reacting to vasoconstrictors. This so-called “arterial paralysis” hypothesis was widely accepted as a mechanism underlying successful angioplasty in the PTCA era [22]. Constrictor resistance (Figure 4), seen in 35.5% of the DCB-treated lesions, supports that the hypothesis may be valid when treating de novo native coronary lesions using DCB. A further study is warranted to evaluate its implication for successful DCB angioplasty and longer-term prognosis. Many studies have shown that post-angioplasty vasospasm is mediated by vasoactive substances released by aggregating platelets (e.g., thromboxane and serotonin) at the site of vessel injury [23,24,25], and aspirin pretreatment can significantly ameliorate the effect [24]. Routine use of both aspirin and P2Y12 inhibitors in contemporary PCI practice could minimize the platelet-mediated deleterious effect. Statin treatment has also been shown to play a role in suppressing coronary spasm by inhibiting vascular smooth muscle contraction [26,27]. The adherence of our study to the current recommendation of lipid-lowering treatment is thought to be another factor contributing to the current findings. Recently, a preclinical study using a novel dual-modal optical coherence tomography and near-infrared fluorescence imaging showed that DCB angioplasty could lessen the inflammation of neoatheroma that develops afterward [28]. Given the suggested mechanistic link between inflammation and coronary vasospasm [29], the anti-inflammatory effect of DCBs may have also contributed to the findings.

This study is limited by a relatively small population. Angiographically normal segments, not evidenced by intravascular imaging, may harbor atherosclerosis. Unlike acetylcholine affecting both the endothelium and vascular smooth muscle, the ergonovine effect primarily represents endothelium-independent constriction [14]. Thus, our findings provide indirect evidence regarding endothelial recovery following DCB angioplasty, although it is conceivable that the ergonovine effect is strongly reinforced in the absence of a functional endothelium [21]. Although this all-comers prospective registry of DCB angioplasty recipients is based on routine 6- to 9-month follow-up coronary angiography, a substantial number of participants, especially those with multiple comorbidities, were ineligible for the provocation test. Therefore, selection bias cannot be excluded.

## 5. Conclusions

Abnormal coronary vasomotion, as invasively assessed using intracoronary vasoactive substances, is an independent prognosticator for adverse cardiovascular events [17,30]. Thus, assessment of vasomotor function provides a measure of the safety and efficacy of a certain revascularization strategy. Our data support that DCB-only PCI for de novo native coronary artery lesions is safe in terms of post-angioplasty spasm risk and allows early restoration of vasomotor function. Further long-term studies are warranted to evaluate whether the observed favorable results translate into reduction of recurrent coronary events and amelioration of atherosclerosis that develops following DCB angioplasty.

## Figures and Tables

**Figure 1 jcm-11-00299-f001:**
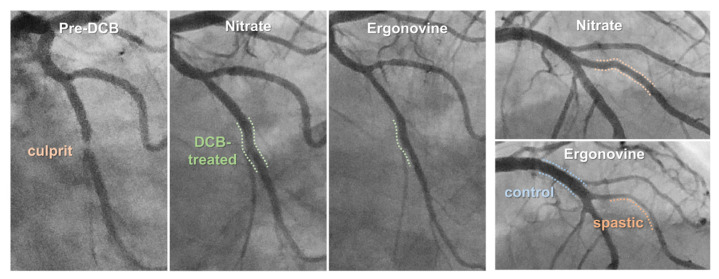
Representative case showing the three comparative arterial segments. In this patient who received DCB angioplasty at left circumflex artery, proximal left anterior descending artery serves as angiographically normal segment and big diagonal branch serves as spastic counterpart. DCB: drug-coated balloon.

**Figure 2 jcm-11-00299-f002:**
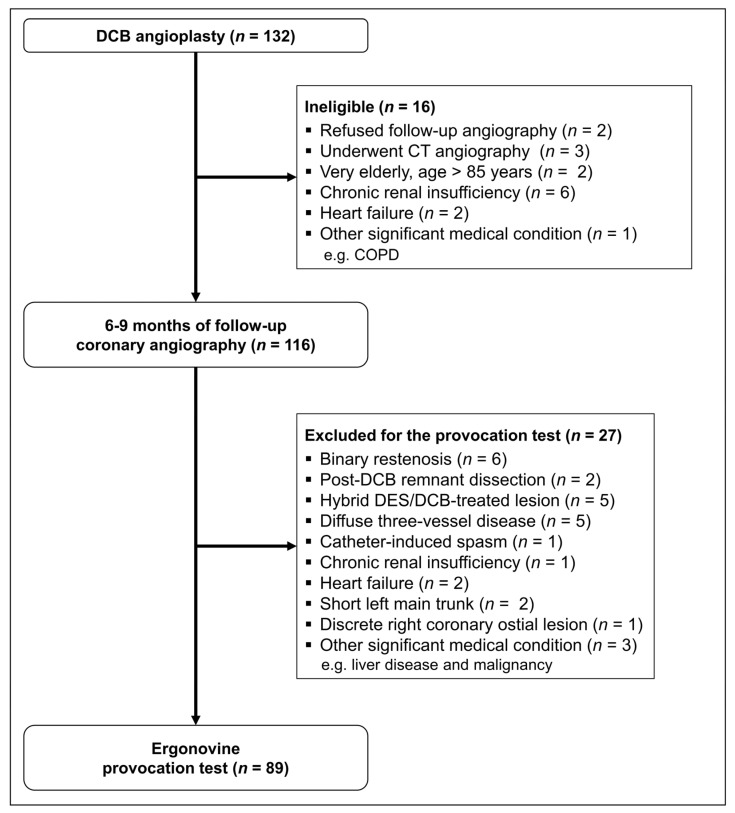
CONSORT flow diagram of this study. CT, computed tomography; COPD, chronic obstructive pulmonary disease; DES, drug-eluting stent; DCB, drug-coated balloon.

**Figure 3 jcm-11-00299-f003:**
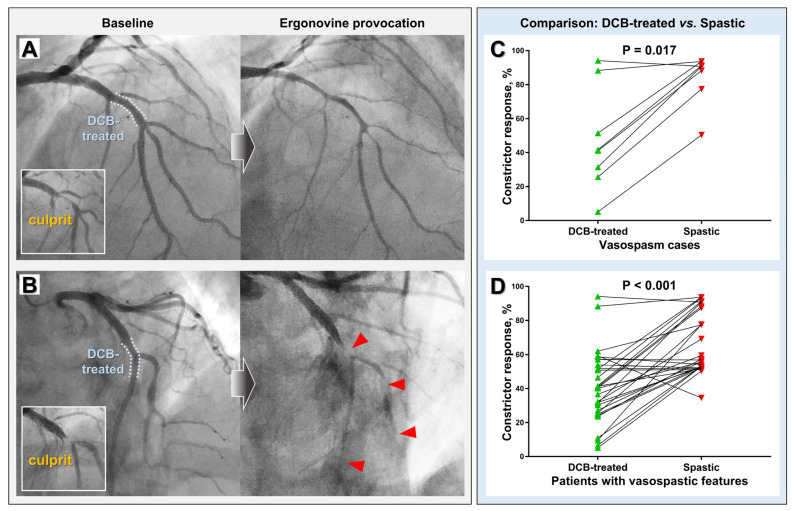
Representative vasospasm cases (**left panel**) and comparison of ergonovine-mediated vasoconstrictive response between DCB-treated and spastic segments (**right panel**). (**A**) A case showing ergonovine-induced spasm that developed diffusely throughout multiple territories while sparing the DCB-treated lesion. (**B**) A case showing multifocal spasm that involved both the DCB-treated lesion and neighboring arterial segments (red arrowheads). Paired comparison results of seven vasospasm cases (**C**) and twenty-nine patients with vasospastic features (**D**). DCB, drug-coated balloon.

**Figure 4 jcm-11-00299-f004:**
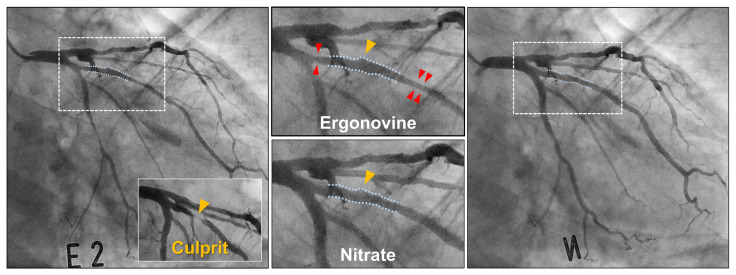
DCB-treated lesion showing constrictor resistance. The DCB-dilated culprit site (yellow arrowheads) is nearly unaffected by ergonovine, while the adjacent proximal and distal portions show vasoconstrictor response (red arrowheads).

**Figure 5 jcm-11-00299-f005:**
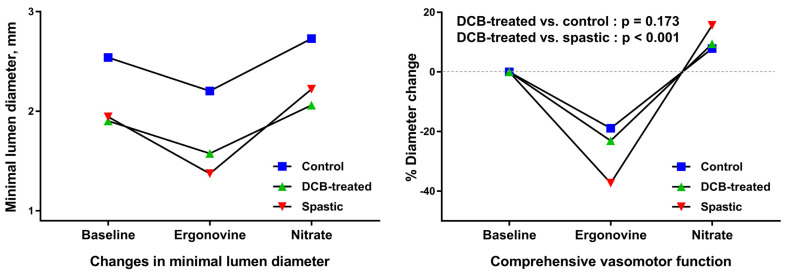
Comparison of serially assessed vasomotor response to ergonovine and nitrate between the three comparative segments.

**Table 1 jcm-11-00299-t001:** Clinical, lesional, and procedural characteristics of 89 patients who underwent the provocation test.

Clinical Characteristics	Patient (*n* = 89)	Lesion Characteristics	Lesion (*n* = 118)
Age (years)	61.8 ± 9.3	Lesion location	
Male	67 (75.3%)	Left anterior descending	39 (33.1%)
Smoking	24 (27.0%)	Left circumflex	30 (25.4%)
Hypertension	54 (60.7%)	Right coronary	21 (17.8%)
Diabetes	32 (36.0%)	Diagonal	10 (8.5%)
Dyslipidemia	62 (69.7%)	Obtuse marginal	9 (7.6%)
Previous PCI	21 (23.6%)	Ramus	4 (3.4%)
Post-PCI medication		PDA or PLV	5 (4.2%)
Aspirin + Clopidogrel	79 (88.8%)	ACC-AHA classification	
Aspirin + Ticagrelor	10 (11.2%)	A	14 (11.9%)
Statin	13 (14.6%)	B1	33 (28.0%)
Statin/Ezetimibe	76 (85.4%)	B2	45 (38.1%)
Calcium channel blocker	43 (48.3%)	C	26 (22.0%)
ACEi	21 (23.6%)	DCB diameter, mm	2.74 ± 0.37
ARB	39 (43.8%)	Final DCB diameter, mm	2.93 ± 0.38
Beta-blocker	58 (65.2%)	DCB length, mm	25.34 ± 4.99
Nitrates	26 (29.2%)	Post-DCB dissection	43 (36.4%)

Values are means ± standard deviation or counts and percentages. PCI: percutaneous coronary intervention; PDA: posterior descending artery; PLV: posterior left ventricular artery; DCB: drug-coated balloon.

**Table 2 jcm-11-00299-t002:** Angiographic characteristics of the analyzed 274 coronary arterial segments.

	Control	DCB-Treated	Spastic	* *p* Value
Reference vessel diameter, after nitrate (mm)	3.12 ± 0.60 ^†^	2.72 ± 0.41	2.70 ± 0.48	0.718
Diameter stenosis, after nitrate (%)	12.72 ± 0.68 ^†^	24.21 ± 0.82	17.57 ± 1.00	<0.001
Minimal lumen diameter, baseline (mm)	2.54 ± 0.57 ^†^	1.90 ± 0.42	1.94 ± 0.46	0.438
Minimal lumen diameter, after ergonovine (mm)	2.20 ± 0.57 ^†^	1.58 ± 0.41	1.37 ± 0.44	<0.001
Minimal lumen diameter, after nitrate (mm)	2.73 ± 0.61 ^†^	2.06 ± 0.40	2.22 ± 0.47	0.003

Values are means ± standard deviation. * *p* value denotes the difference between DCB-treated vs. spastic segments. ^†^ Significantly different compared to both DCB-treated and spastic segments (*p* < 0.001). DCB: drug-coated balloon.

## Data Availability

The data presented in this study are available on a reasonable request from the corresponding author. More data are contained within the Appendix A.

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
