# Peer review of "Favorable Vasomotor Function after Drug-Coated Balloon-Only Angioplasty of De Novo Native Coronary Artery Lesions"

_jcm, 2022, doi:10.3390/jcm11020299_

Round 1
Reviewer 1 Report
Sunwon Kim et al. present a well conducted prospective single center study evaluating all patients with significant CAD who received PCI with DCB-only angioplasty, concomitant DCB angioplasty and POBA, or concomitant DCB angioplasty and DES implantation for de novo lesions in the native coronary arteries.
The main question addressed by the researchers was the effect of DCB on vasomotor properties of coronary arteries especially analyzing vasospasm and the respond to ergonovine and nitrate
The topic is interesting as POBA is known to alter vessel vasomotility as it is mainly used to predilate coronary lesions in order to prepare the vessel for stent implant while DCB are gently inflanted to elute antiproliferative drugs to the vessels
The paper is original and novel as it shows that DCB-treated lesions were not particularly vulnerable to vasospasm and that mostly were found to have vasomotor function similar to angiographically normal segments, supporting the safety of DCB-only strategy in treating de novo native coronary lesions.
The paper is well written and easy to understand with conclusions consistent with the presented results
Minor comment: I would better clarify the title with this change: “Favorable Vasomotor Function after Drug-Coated Balloon-Only Angioplasty of De Novo Coronary Arteries Lesions”
Author Response
We are very grateful to the reviewers for the meticulous examination and valuable comment on our manuscript. The title of this article is now revised as recommended.

Reviewer 2 Report
Dear authors:
1.- Please provide AHA classification of the lesions, which could be included in table 1.
2.- Line 90 : Change "sufficient dose of intracoronary isosorbide dinitrate" for a real dose (or a range of dose that you usually use in these terms)
3.-Line 134: DS , please write Diametre Stenosis (DS)
4.- Line 137: RVD., please write Reference Vessel Diameter (RVD)
5.- Material and methods: Please provide % of use of Scoring balloon or plain balloon.
6.- Table 1:
Please provide the name and % of antiplatelets Post PCI medication.
From Line 232 to 238: Values must be included in Table 2.
Author Response
We are very grateful to the reviewers for the meticulous examination and valuable comments on our manuscript. Our responses to the reviewers’ comments are listed in the attached file. Please see the attachment.
